**Data Availability Statement:** All relevant data are within the manuscript and its Supporting Information files.

**Funding:** Young-Hoon Park 2016R1A6A1A03010528 Basic Science Research

# Prognostic factors of cytomegalovirus retinitis after hematopoietic stem cell transplantation

**Joo Young Kim**[1,2], **Seo-Yeon Hong**[1], **Woo Kyung Park**[1,2], **Rae Young Kim**[1,2], **Mirinae Kim**[1,2], **Young-Gun Park**[1,2], **Hee-Je Kim**[3], **Seok Lee**[3], **Dong Gun Lee**[4], **Young-Hoon Park**[1,2]*

**1** Department of Ophthalmology and Visual Science, Seoul St. Mary's Hospital, College of Medicine, The Catholic University of Korea, Seoul, Korea, **2** Catholic Institute for Visual Science, College of Medicine, The Catholic University of Korea, Seoul, Korea, **3** Catholic Hematology Hospital, Leukemia Research Institute, Seoul St. Mary's Hospital, College of Medicine, The Catholic University of Korea, Seoul, Korea, **4** Division of Infectious Disease, Department of Internal Medicine, Seoul St. Mary's Hospital, College of Medicine, The Catholic University of Korea, Seoul, Korea

* parkyh@catholic.ac.kr

## Abstract

### Purpose

To identify the visual prognostic factors in patients with cytomegalovirus (CMV) retinitis after hematopoietic stem cell transplantation (HSCT).

### Methods

This retrospective cohort study included 4241 patients who underwent HSCT from April 1, 2010 to March 31, 2019 at Seoul St. Mary's Hospital. Of them, 1063 patients presented CMV viremia, and 67 patients (93 eyes) were diagnosed with CMV retinitis. We enrolled 66 patients (91 eyes). The main outcomes included the initial best-corrected visual acuity (BCVA), BCVA at the diagnosis of retinitis and last visit, involved retinal zone, peak CMV DNA levels in the peripheral blood and aqueous humor, time between HSCT and the diagnosis of retinitis, time between the diagnosis of viremia and retinitis, complications, recurrence, survival, and so on.

### Results

The mean BCVA (logarithm of the minimum angle of resolution) values before HSCT, at the time of retinitis diagnosis, and at the last visit were 0.041 ± 0.076, 0.262 ± 0.529, and 0.309 ± 0.547, respectively. Multiple regression analysis revealed that the involved zone (P = 0.001), time between HSCT and retinitis diagnosis (P = 0.019), and survival status (P = 0.001) were associated with the final visual acuity.

### Conclusions

The final visual prognosis was worse in patients with greater invasion of the central retinal zone, those with a longer interval between HSCT and the diagnosis of retinitis, and those who died. Prompt diagnosis of CMV retinitis through periodic fundus examinations of

Program through the National Research Foundation of Korea (NRF) The funders had no role in study design, data collection and analysis, decision to publish, or preparation of the manuscript.

**Competing interests:** The authors have declared that no competing interests exist.

patients with CMV viremia can prevent severe vision loss. Once CMV viremia is confirmed, we recommend fundus examinations to be immediately performed and repeated every 2 weeks for at least 2 months, even if the CMV DNA titer in the peripheral blood becomes negative.

## Introduction

Cytomegalovirus (CMV) retinitis is an opportunistic infection that occurs in immunocompromised patients such as those with AIDS, hematopoietic stem cell transplantation (HSCT), solid organ transplantation, malignancy, systemic immunosuppressive therapy, and primary immunodeficiency. This form of retinitis can lead to severe vision loss if it is not promptly diagnosed and appropriately treated [1–5].

CMV remains in a latent state in the myeloid progenitor cells in the bone marrow or in a low-level replication state, facilitated by multiple immune evasion mechanisms. When it is intermittently stimulated, it releases virions and is reactivated, although the condition is mostly asymptomatic and self-limiting. However, CMV shedding may be dramatically increased in seropositive patients with impaired immune systems [6]. When CMV DNA exits the bone marrow through the myeloid progenitor cells and enters the peripheral blood, macrophages and dendritic cells resulting from differentiation of the progenitor cells are believed to reactivate the virus [7]. CMV reactivation can result in retinitis, hepatitis, esophagitis, gastroenteritis, colitis, pneumonitis, encephalitis, and polyradiculopathy [8–10].

Features of CMV retinitis in HSCT recipients differ from those in patients with other causes of immunodeficiency. First, HSCT recipients require more amount of virus than AIDS patients to cause CMV retinitis. HIV infects the retinal pigment epithelium (RPE) and disrupts the outer blood-retina barrier (BRB), making it easier for CMV to invade the inner retina in AIDS patients [3, 11, 12]. For this reason, the incidence of CMV retinitis in AIDS patients reached 30% before highly-active antiretroviral therapy (HAART) was introduced [5, 13, 14]. In the HAART era, CMV retinitis was significantly reduced [15], but it is still the most common form of CMV disease in AIDS patients [6]. However, it is not the most common form of CMV disease in HSCT patients [16], and the incidence is reported as 0.17–2.3% [3, 17–20]. Second, the greatest risk factor for CMV infection after solid organ transplantation is a CMV-seropositive donor to a CMV-seronegative recipient. This is because the recipients lack the T-cells required to fight CMV infection. Conversely, the greatest risk factor for CMV infection after allogeneic HSCT is transplantation from CMV-seronegative donors to CMV-seropositive recipients. Prior to transplantation, chemotherapy not only eradicates leukemia/lymphoma but also eliminates the immune system of recipients in order to prevent graft-versus-host disease (GVHD). Then the donor's immune system and T-cells, which have never been exposed to CMV, replaces the host's immune system [1, 3, 10, 21].

Previous studies have shown high-risk characteristics for the development of CMV retinitis like prolonged CMV viremia, high peak CMV DNA levels in the peripheral blood after HSCT, transplants from unrelated donors, and transplants from human leukocyte antigen (HLA)-mismatched donors [3]. However, factors affecting the visual prognosis of patients diagnosed with CMV retinitis after HSCT remain unclear. Accordingly, the aim of the present study was to investigate the visual prognostic factors in patients diagnosed with CMV retinitis after HSCT.

## Methods

### Patients

This retrospective cohort study included patients diagnosed with CMV retinitis among those presenting with CMV viremia after HSCT. From April 1, 2010 to March 31, 2019, 4241 patients underwent HSCT at Seoul Saint Mary's Hospital. From these, 1063 patients presented CMV viremia, which manifested as CMV retinitis in 67 patients (93 eyes). All patients diagnosed with CMV retinitis were of Korean descent. One patient with bilateral vision loss due to infectious endophthalmitis before the diagnosis of CMV retinitis was excluded. Eventually, we enrolled 66 patients (91 eyes) and retrospectively analyzed their medical records and laboratory reports. Institutional Review Board/Ethics Committee approval was obtained (from Seoul Saint Mary's Hospital), and the study was conducted in accordance with the Declaration of Helsinki. The need for informed consent was waived due to the retrospective nature of the study.

### Diagnosis

All patients scheduled for HSCT underwent best-corrected visual acuity (BCVA) and intraocular pressure measurements, slit-lamp microscopy, and fundus examination before HSCT. We measured CMV DNA levels in peripheral venous blood samples using quantitative polymerase chain reaction (PCR) before and once a week for 3 months after HSCT. Fundus examinations were performed as soon as patients were diagnosed with CMV viremia, and they were performed such that the patients' general condition was not affected. If CMV viremia was not detected, without ophthalmic symptoms, no additional examination was performed. Patients with deteriorated systemic conditions occasionally received bedside examinations; however, fundus examinations were delayed for patients with an extremely poor general condition. If CMV viremia persisted without the onset of the clinical manifestations of CMV retinitis, we repeated the fundus examinations after 2 weeks. Even if CMV was no longer detected in the peripheral blood, fundus examinations were repeated every 2 weeks for 2 months. After 2 months, the interval was gradually increased, and the examination was discontinued. Fundus examination was then performed only when CMV viremia appeared again or ophthalmic symptoms occurred.

A retinal specialist diagnosed CMV retinitis when a pale necrotic retina with focal areas of hemorrhage or variable, small dot-like lesions, or retinal vasculitis with perivascular sheathing starting from the peripheral retina and centrifugally progressing toward the posterior pole, were observed [1, 21]. In addition, we performed CMV DNA PCR using aqueous humor samples obtained by anterior chamber paracentesis in all patients showing clinical manifestations of CMV retinitis. Samples were collected under aseptic conditions in the ophthalmology clinic. This test was also performed when recurrence or aggravation was suspected or the treatment response was evaluated.

We divided the retina into zones 1, 2, and 3 in order to distinguish the extent of invasion by CMV. Zone 1 represents an area of up to 3000 μm (2 disc diameters) from the fovea or 1500 μm from the edge of the optic nerve head. Zone 2 represents an area extending forward from zone 1 to the clinical equator of the eye, and zone 3 represents an area extending forward from zone 2 to the ora serrata [22].

### Treatment

All HSCT recipients were administered with intravenous tacrolimus (Prograf, Astellas, Tokyo, Japan) or cyclosporine (Sandimmun, Novartis, Basel, Switzerland) for the prevention of

GVHD. On detecting CMV viremia, we initiated intravenous infusion of ganciclovir (Cymevene, Hoffmann-La Roche, Basel, Switzerland) according to the protocol for risk-adapted preemptive therapy for CMV disease after HSCT [23]. However, in cases of severe neutropenia, anemia, or thrombocytopenia, we administered intravenous foscarnet (Foscavir, Fresenius Kabi, Graz, Austria) instead of ganciclovir. Oral valganciclovir (Valcyte, Hoffmann-La Roche) was used for maintenance therapy [24].

All patients with CMV retinitis initially received an intravitreal injection of ganciclovir. In cases of ganciclovir resistance, we administered intravitreal foscarnet [25, 26]. Patients with reduced CMV viremia who required further systemic treatment for CMV retinitis received the necessary treatment.

## Analysis

The following data were collected from the medical records of the included patients: the patient's age and sex, involved eye, initial BCVA (before HSCT), BCVA at the diagnosis of CMV retinitis, BCVA at the last visit, ophthalmic symptoms at diagnosis, involved retinal zone, hematological disease, type of HSCT (HLA-matched vs. mismatched donor, related vs. unrelated donor), peak CMV DNA levels detected in the peripheral blood and aqueous humor, systemic use of ganciclovir/foscarnet/valganciclovir, intravitreal injection of ganciclovir/foscarnet, recurrence of retinitis, survival status, and complications such as rhegmatogenous retinal detachment (RRD) and immune recovery uveitis (IRU).

BCVA values were converted into a logarithm of the minimal angle of resolution (logMAR) units. Patients who could only count fingers or detect hand motion were assigned logMAR values of 2.0 and 3.0, respectively. Cases of light perception and no light perception were excluded from the calculation for the mean logMAR BCVA [27].

All data were statistically analyzed using the Statistical Package for the Social Sciences software, version 24.0 (SPSS Inc, Chicago, IL, USA). Multiple regression analysis and Pearson's correlation analysis were performed for prognostic factors affecting the final visual acuity. The survival rate according to the follow-up period was evaluated by Kaplan-Meier survival analysis. A log-rank test was used to compare the survival rates of the 2 groups divided according to final visual acuity. A $P$-value of $<0.05$ was considered statistically significant.

## Results

In total, 91 eyes of 66 patients diagnosed with CMV retinitis were included. CMV DNA was detected in the aqueous humor of all patients diagnosed with CMV retinitis. All participants had CMV viremia either at the time retinitis was diagnosed, or at some time previously. The patients' demographic characteristics are described in Table 1.

The mean logMAR BCVA before HSCT, at the time of retinitis diagnosis, and at the last visit were 0.041 ± 0.076 (range in Snellen, 20/40-20/20), 0.262 ± 0.529 (range in Snellen, hand motion-20/20), and 0.309 ± 0.547 (range in Snellen, no light perception-20/20), respectively; these values were calculated after excluding 2 patients with a final visual acuity of no light perception in both eyes.

Only 44 eyes (48.4%) exhibited ophthalmic symptoms such as decreased visual acuity, blurred vision, floaters, and, rarely, visual field defects. Thirteen eyes (14.3%) of 10 patients (15.2%) were diagnosed with CMV retinitis after negative peripheral blood CMV testing. The average duration between the detection of CMV negativity and the diagnosis of retinitis was 24.64 ± 16.04 (7–54) days.

RRD occurred in 7 eyes (7.7%) of 7 patients (10.6%), 4 of whom were diagnosed with unilateral CMV retinitis. Three patients with bilateral retinitis exhibited monocular RRD. IRU

**Table 1. Characteristics of patients with cytomegalovirus retinitis after hematopoietic stem cell transplantation.**

|  | Total (n = 66) |
|---|---|
| Sex | 33:33 |
| Male:Female | |
| Age (years), median (range) | 43.0 (16.0–68.0) |
| Hematological disease | |
| Acute lymphoblastic leukemia | 33 |
| Acute myeloid leukemia | 14 |
| T-cell lymphoma | 7 |
| Aplastic anemia | 4 |
| Diffuse large B-cell lymphoma | 3 |
| Myelodysplastic syndrome | 3 |
| Multiple myeloma | 1 |
| Primary myelofibrosis | 1 |
| Donor HLA matching | 29:37 |
| matched:mismatched | |
| Donor relation | 33:33 |
| related:unrelated | |

HLA, human leukocyte antigen.

occurred in 14 eyes (15.4%) of 10 patients (15.2%). Five eyes (5.5%) of 4 patients (6.1%) developed macular edema secondary to IRU.

Fifteen eyes (16.5%) of 11 patients (16.7%) exhibited recurrence of retinitis after the retinal lesion had disappeared and the aqueous humor tested negative for CMV. From these, 5 patients had unilateral retinitis with unilateral recurrence, 2 had bilateral retinitis with unilateral recurrence, and 4 had bilateral retinitis with bilateral recurrence. In these cases, treatment with ganciclovir was continued if the initial treatment had been effective.

Twenty patients (30.3%) died because of a worsening systemic condition within 1 year after the diagnosis of retinitis. Ten patients died while receiving treatment for retinitis, while 10 did not receive ophthalmic treatment because they exhibited CMV negativity in the peripheral blood and amelioration of retinitis. The time of period from the last visit to patients' death was between 3 days and 3 months. The remaining 46 patients were followed-up for more than 1 year after the resolution of CMV retinitis (Table 2). Treatment outcomes for patients with CMV retinitis are described in Table 3.

According to Pearson's correlation analysis, the involved zone ($P = 0.008$), the time between HSCT and the diagnosis of CMV retinitis ($P = 0.004$), and survival status ($P = 0.002$) were significant factors influencing the final visual acuity. These factors were also significant in multiple regression analyses ($P = 0.001$, 0.019, and 0.001, respectively). Greater invasion of the central retina and a longer interval between HSCT and the diagnosis of CMV retinitis were associated with a worse final visual acuity. Moreover, the final visual acuity was worse in patients who died than in those who survived. In addition, Kaplan-Meier survival analysis showed a difference in survival rate between the 2 groups divided according to final visual acuity; those with poor final visual acuity had a lower overall survival rate ($P = 0.013$, log-rank test) (Fig 1). Other clinical factors, including the peak CMV DNA levels in the peripheral blood and aqueous humor, time between the diagnosis of CMV viremia and the diagnosis of CMV retinitis, treatment duration for CMV retinitis, number of intravitreal antiviral injections, recurrence of CMV retinitis, donor HLA matching, and donor relation showed no significant correlations with the final

**Table 2. Clinical outcomes of patients with cytomegalovirus retinitis after hematopoietic stem cell transplantation.**

| | According to the number of eyes (n = 91) | According to the number of patients (n = 66) |
|---|---|---|
| Involved eye | | |
| Both eyes | | 25 (37.9%) |
| Right eye | | 22 (33.3%) |
| Left eye | | 19 (28.8%) |
| logMAR BCVA (range in Snellen) | | |
| at the initial visit | 0.041 ± 0.076 (20/40-20/20) | |
| at the diagnosis of CMV retinitis | 0.262 ± 0.529 (HM-20/20) | |
| at the last visit | 0.309 ± 0.547[a] (NLP-20/20) | |
| BCVA (Snellen), > 20/40 | 59 (64.8%) | |
| BCVA (Snellen), ≤ 20/40 | 19 (20.9%) | |
| BCVA (Snellen), ≤ 20/200 | 13 (14.3%) | |
| Ophthalmic symptoms at diagnosis | | |
| Present | 44 (48.4%) | |
| Absent | 47 (51.6%) | |
| Involved retinal zone | | |
| Zone 1 | 31 (34.1%) | |
| Optic disc involvement | 4 (4.4%) | |
| Fovea involvement | 9 (9.9%) | |
| Optic disc & fovea involvement | 6 (6.6%) | |
| Zone 2 | 33 (36.3%) | |
| Zone 3 | 27 (29.7%) | |
| Patients diagnosed with CMV retinitis after the peripheral blood tested negative for CMV DNA[b] | 13 (14.3%) | 10 (15.2%) |
| Rhegmatogenous retinal detachment | 7 (7.7%) | 7 (10.6%) |
| Immune recovery uveitis | 14 (15.4%) | 10 (15.2%) |
| Macular edema | 5 (5.5%) | 4 (6.1%) |
| Recurrence of CMV retinitis | 15 (16.5%) | 11 (16.7%) |
| Death within 1 year after the diagnosis of retinitis | 27 (29.7%) | 20 (30.3%) |
| last visit logMAR BCVA (range in Snellen) | | |
| patients alive | 0.271 ± 0.511 (HM-20/20) | |
| patients dead | 0.429 ± 0.631[a] (NLP-20/20) | |

Values are presented as mean ± standard deviation unless otherwise indicated.

[a]Two patients with bilateral visual acuity with NLP were excluded from the calculation for the mean BCVA, because NLP cannot be expressed as measurements such as logMAR [27].

[b]Ten patients (13 eyes) were diagnosed with CMV retinitis at an average of 24.64 ± 16.04 (range, 7–56) days after the CMV DNA titer in the peripheral blood became negative.

BCVA, best-corrected visual acuity; logMAR, logarithm of the minimal angle of resolution; CMV, cytomegalovirus; HM, hand motion; NLP, no light perception.

visual acuity (Table 4). The clinical features and treatment outcomes of patients with severe complications of CMV retinitis are described in Table 5.

## Discussion

In the present study, 1063 of 4241 HSCT recipients (25.06%) were diagnosed with CMV viremia, with 67 (1.58%) developing CMV retinitis. In a previous study by Jeon et al. [3], 363 of 708 HSCT recipients (51.27%) were diagnosed with CMV viremia, with 15 (2.12%) developing CMV retinitis. In previous studies by Xhaard et al. [17], Crippa et al. [18], and Yan et al. [19],

**Table 3. Treatment outcomes for patients with cytomegalovirus retinitis after hematopoietic stem cell transplantation.**

| | Median (range) | Mean ± standard deviation |
|---|---|---|
| Peak CMV DNA levels in the peripheral blood (copies/ml) | 32500 (500–3904037) | 233749.32 ± 699194.35 |
| Peak CMV DNA levels in the aqueous humor (copies/ml) | 11430 (500–12500000) | 379056.05 ± 1414190.90 |
| Time between HSCT and the diagnosis of CMV retinitis (days) | 75 (28–309) | 94.96 ± 67.21 |
| Time between the diagnosis of CMV viremia and the diagnosis of CMV retinitis (days) | 62 (6–230) | 70.82 ± 59.28 |
| Treatment duration for CMV retinitis (days) | 56 (7–455) | 89.77 ± 86.92 |
| Number of intravitreal antiviral injections | 7 (1–28) | 9.02 ± 6.75 |
| Follow-up period from diagnosis of CMV retinitis to last visit (months) | 13.93 (0.70–104.43) | 23.08 ± 25.20 |

Values are presented as mean ± standard deviation unless otherwise indicated.

CMV, cytomegalovirus; HSCT, hematopoietic stem cell transplantation.

the incidences of CMV retinitis after HSCT were 1.92%, 0.17%, and 2.3%, respectively. Several studies have shown that CMV seropositivity in HSCT recipients is an important risk factor for CMV viremia [10, 21]. Koreans exhibit a high incidence of CMV retinitis because the majority exhibit CMV seropositivity [3]. CMV seroprevalence is about 83% worldwide [28], but 94% in Korea [29].

In the study by Crippa et al. [18] and Yan et al. [19], the median time between the diagnosis of CMV retinitis and HSCT was 251 (range, 106–365) and 167 (range, 60–445) days, respectively; this was considerably higher than the median interval of 75 (range, 28–309; average, 94.96 ± 67.21) days in the present study. The patients in this study (with the exception of some whose fundus examinations were delayed due to deteriorated general condition), received immediate and 2-weekly examinations after CMV viremia was diagnosed. This could have shortened the time until diagnosis of CMV retinitis.

Despite adequate systemic treatment for CMV viremia, 10 patients were diagnosed with CMV retinitis after an average of 24.64 ± 16.04 (range, 7–56) days, after CMV DNA titer in the

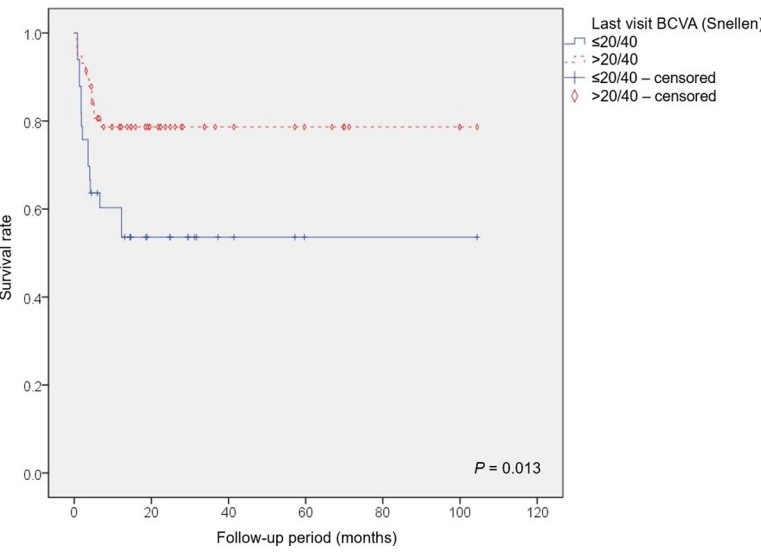

**Fig 1. Kaplan-Meier survival analysis showed a difference in survival rate over the follow-up period between the 2 groups, which were divided according to final best-corrected visual acuity (BCVA).** Of the 59 eyes with a final BCVA >20/40, 12 eyes (9 patients) died, and 15 eyes (11 patients) of 32 eyes with a final BCVA ≤20/40 died. Patients with poor final BCVA had a lower overall survival rate ($P$ = 0.013, log-rank test).

**Table 4. Prognostic factors affecting the final visual acuity of cytomegalovirus retinitis occurring after hematopoietic stem cell transplantation.**

| Independent Variables | Pearson's Correlation Analysis | | Multiple Regression Analysis | | | |
|---|---|---|---|---|---|---|
| | Coefficient | P-value | Unstandardized Beta Coefficient | Standardized Beta Coefficient | t | P-value |
| Peak CMV DNA levels in the peripheral blood (copies/ml) | −0.067 | 0.528 | −2.097E-7 | −0.108 | −1.192 | 0.237 |
| Peak CMV DNA levels in the aqueous humor (copies/ml) | 0.038 | 0.730 | −5.928E-8 | −0.008 | −0.092 | 0.927 |
| Involved retinal zone | −0.275 | 0.008** | −0.491 | −0.422 | −4.773 | 0001** |
| Time between HSCT and the diagnosis of CMV retinitis (days) | 0.303 | 0.004** | 0.003 | 0.209 | 2.380 | 0.019* |
| Time between the diagnosis of CMV viremia and the diagnosis of CMV retinitis (days) | 0.186 | 0.077 | 0.003 | 0.040 | 0.441 | 0.660 |
| Treatment duration for CMV retinitis (days) | 0.124 | 0.241 | 0.001 | 0.114 | 1.111 | 0.270 |
| Number of intravitreal antiviral injections | 0.204 | 0.052 | 0.030 | 0.170 | 1.818 | 0.073 |
| Recurrence of CMV retinitis | 0.066 | 0.531 | 0.332 | 0.056 | 0.601 | 0.549 |
| Donor HLA matching | −0.169 | 0.108 | 0.022 | 0.036 | 0.392 | 0.696 |
| matched:mismatched | | | | | | |
| Donor relation | 0.182 | 0.084 | 0.362 | 0.153 | 1.729 | 0.087 |
| related:unrelated | | | | | | |
| Survival:Death | −0.334 | 0.002** | −0.786 | −0.387 | −4.343 | 0.001** |
| | | | Durbin–Watson = 1.537, $R^2$ = 0.330, F = 4.538** (<0.001) | | | |

Correlations are significant at the 0.05 level.

*$P < 0.05$,

**$P < 0.01$.

CMV, cytomegalovirus; HSCT, hematopoietic stem cell transplantation; HLA, human leukocyte antigen.

peripheral blood became negative, probably because CMV takes a long time to infect the retina through BRB. Yanping et al. [30] suggested that murine CMV retinitis requires more than 2 weeks to develop, and Gao et al. [31] showed that murine intraocular CMV replication is independent of systemic CMV replication and occurs after a certain time interval.

Although periodic retinal screening after HSCT is expensive, time-consuming, and uncomfortable to the patient, CMV retinitis can be diagnosed in many patients despite the absence of ophthalmic symptoms. The patients with a shorter interval between HSCT and diagnosis of CMV retinitis exhibited better visual prognosis; therefore, once CMV viremia is confirmed, we recommend an immediate fundus examination. While CMV viremia is persistent, it is repeated every 2 weeks even if there is no CMV retinitis. We also recommend repeated fundus examinations every 2 weeks for at least 2 months, even after CMV DNA titers in the peripheral blood become negative. The minimum period of up to 2 months is because of the 10 patients (13 eyes) that were diagnosed with CMV retinitis even after 24.64 ± 16.04 (7–56) days after CMV DNA became negative in the peripheral blood. Larochelle et al. [32] designed a CMV retinitis screening protocol for pediatric HSCT recipients. They suggested that all HSCT recipients that are confirmed with CMV viremia or any with CMV organ disease undergo a fundus examination within 2 weeks of viremia being diagnosed, and every 6–8 weeks until no viremia. However, based on our experience, it can be risky to discontinue fundus examinations once blood samples' CMV negativity is achieved. Further research is needed to determine the usefulness of the screening protocol.

In the present study, the patients who died, presented a poor final visual acuity, similar to findings in previous studies of patients with CMV retinitis and AIDS. In 2 previous studies conducted by Jabs et al., patients who experienced immune recovery exhibited a high survival

**Table 5. Clinical characteristics and outcomes of patients with severe complications of cytomegalovirus retinitis that occurred after hematopoietic stem cell transplantation.**

| Sex/ Age (years) | Diagnosis | Involved eye | Involved retinal zone | Intravitreal antiviral injections | | Complications | | Time between HSCT and the diagnosis of retinitis (days) | Peak CMV DNA levels in peripheral blood (copies/ml) | Peak CMV DNA levels in aqueous humor (copies/ml) | BCVA (Snellen) | | | Death |
| --- | --- | --- | --- | --- | --- | --- | --- | --- | --- | --- | --- | --- | --- | --- |
| | | | | Medication | Number | Type | Treatment | | | | Before HSCT | At the diagnosis of retinitis | At the last visit | |
| M/47 | ALL | OD | 2 | GCV | 5 | RRD | PPV | 122 | 2722778 | 460394 | 20/20 | 20/20 | 20/64 | No |
| | | OS | 2 | GCV | 5 | | | 122 | | 29405 | 20/20 | 20/20 | 20/32 | |
| F/48 | AA | OD | 1 | GCV | 12 | VO | | 161 | 123000 | 39300 | 20/25 | 20/50 | 20/50 | No |
| | | OS | 1 | GCV | 12 | RRD, VO | Encircling +PPV | 161 | | 268000 | 20/40 | 20/200 | 20/250 | |
| F/39 | ALL | OD | 2 | GCV | 21 | IRU, VO | IVTA | 30 | 566 | 501000 | 20/20 | 20/20 | 20/32 | No |
| | | OS | 1 | GCV | 25 | RRD, IRU, ME | PPV, IVTA | 30 | | 18500 | 20/20 | 20/160 | 20/250 | |
| M/40 | AML | OS | 1 | GCV | 6 | RRD, IRU | Encircling +PPV, SBTA | 51 | 140442 | 1473 | 20/25 | 20/25 | 20/400 | No |
| M/64 | MDS | OD | 2 | GCV | 11 | RRD | PPV | 73 | 35700 | 131000 | 20/20 | 20/25 | 20/25 | No |
| F/66 | AML | OS | 2 | GCV | 9 | RRD, IRU | PPV, IVTA | 54 | 94900 | 31800 | 20/20 | 20/25 | 20/20 | No |
| F/38 | ALL | OD | 1 | GCV | 3 | RRD | Barrier laser | 142 | 3970 | 125000 | 20/20 | 20/20 | 20/20 | No |
| F/55 | ALL | OD | 1 | GCV, FOS | 12, 3 | VH, IRU | PPV, SBTA | 209 | 500 | 1010355 | 20/40 | FC | 20/200 | No |
| M/29 | ALL | OD | 1 | GCV | 24 | VO, IRU | IVTA | 33 | 1183 | 20814 | 20/20 | 20/40 | 20/50 | No |
| F/65 | AML | OD | 1 | GCV | 12 | VO | PPV | 82 | 32500 | 5330 | 20/25 | 20/250 | 20/50 | No |
| F/25 | ALL | OD | 1 | GCV, FOS | 8, 19 | IRU, ME | IVTA | 35 | 3810 | 1760000 | 20/25 | 20/25 | HM | No |
| | | OS | 3 | GCV, FOS | 6, 13 | IRU | IVTA | 35 | | 625000 | 20/20 | 20/20 | 20/25 | |
| M/58 | AML | OS | 2 | GCV | 12 | IRU | SBTA | 54 | 2236 | 6116 | 20/20 | 20/20 | 20/40 | No |
| M/52 | AML | OS | 2 | GCV | 2 | IRU, ME | IVTA | 99 | 93600 | 66800 | 20/20 | 20/20 | 20/40 | No |
| M/43 | ALL | OD | 2 | GCV | 24 | IRU, ME | IVTA | 103 | 10330 | 3871319 | 20/20 | 20/40 | 20/40 | No |
| | | OS | 1 | GCV | 20 | IRU, ME | IVTA | 103 | | 2392813 | 20/20 | 20/40 | 20/32 | |
| F/51 | MDS | OS | 1 | GCV | 16 | Macular scar | | 155 | 47600 | 1850 | 20/25 | 20/100 | FC | No |
| M/61 | DLBL | OD | 1 | GCV, FOS | 20, 1 | VO, IRU | IVTA | 252 | 2710 | 45600 | 20/25 | 20/40 | NLP | Yes |
| | | OS | 1 | GCV, FOS | 20, 1 | VO, IRU | IVTA | 252 | | 38900 | 20/25 | 20/400 | NLP | |
| M/37 | DLBL | OD | 1 | GCV | 3 | | | 148 | 222876 | 757671 | 20/20 | HM | NLP | Yes |
| | | OS | 1 | GCV | 3 | | | 148 | | 1415525 | 20/20 | HM | NLP | |

HSCT, hematopoietic stem cell transplantation; CMV, cytomegalovirus; BCVA, best-corrected visual acuity; ALL, acute lymphoblastic leukemia; AA, aplastic anemia; AML, acute myeloid leukemia; MDS, myelodysplastic syndromes; DLBL, diffuse large B-cell lymphoma; OD, right eye; OS, left eye; GCV, ganciclovir; FOS, foscarnet; RRD, rhegmatogenous retinal detachment; VO, vitreous opacity; IRU, immune recovery uveitis; ME, macular edema; VH, vitreous hemorrhage; PPV, pars plana vitrectomy; IVTA, intravitreal triamcinolone acetonide; SBTA, subtenon triamcinolone acetonide; FC, finger count; HM, hand motion; NLP, no light perception.

rate, whereas those without immune recovery exhibited a low survival rate. In their first study [14], the rates of retinitis progression, second eye involvement, and decreased visual acuity were significantly higher in the no immune recovery group than in the immune recovery group. In the second study [13], however, visual impairment was independent of the immune status. According to the study of Holland et al. [32], in AIDS patients, CMV activity increases in the presence of immunodeficiency. In addition, in the study of Yan et al. [19], patients with CMV retinitis had significantly fewer median absolute counts of lymphocytes, CD3+ cells, and CD8+ T cells than those without CMV retinitis after haploidentical HSCT. The most common cause of death in HSCT recipients was a relapse, infection, and GVHD [33]. In the present study also; the most common causes of death in 20 patients were GVHD and infection. Four patients died of CMV infection in other organs, and 10 died while receiving treatment for CMV retinitis. Control of life-threatening infections, including CMV infection, may be difficult in patients with immunodeficiency; therefore, the final visual acuity of CMV retinitis is likely to be poor.

In the present study, the CMV DNA level in the aqueous humor was used for the diagnosis of CMV retinitis in addition to the characteristic clinical features. In the study by Smith et al., CMV DNA levels in the aqueous humor correlated with the activity and area of retinitis [34]. However, in this study, CMV DNA levels in the peripheral blood, as well as aqueous humor, were not related to the final visual acuity. CMV retinitis usually extends from the periphery to the center. Although the CMV DNA level in the aqueous humor reflects CMV activity and the area of retinitis at the time of measurement, the visual prognosis is determined by the extent of central involvement. If the lesion is confined to the periphery, there may be no correlation between the area affected by retinitis and the visual prognosis. In patients with CMV retinitis, elevated CMV DNA levels in the peripheral blood may indicate extraocular CMV infection or reactivation of CMV retinitis [35], while elevated CMV DNA levels in the aqueous humor may indicate CMV activity in the eye [34]. Therefore, CMV DNA levels can be used to determine recurrence even if it is not related to visual acuity.

This study has some limitations. First, unlike the CMV DNA level in the peripheral blood, which was measured weekly, the CMV DNA level in the aqueous humor was measured only at the time of diagnosis, aggravation, recurrence, and at the end of treatment; this was because aqueous humor sampling is an invasive procedure. Second, it was difficult to determine the precise timing for the onset of immunosuppression, considering HSCT recipients are already immunocompromised because of the underlying hematological disease and the chemotherapy administered before HSCT. Third, the study included many patients in poor general condition, and in some cases, there was no choice but to limit examination or treatment. Some patients had a delayed fundus examination due to worsening of overall condition, and some patients were moved to an intensive care unit during retinitis treatment, so the intravitreal injection was postponed. All patients could undergo visual acuity testing, even by finger counting or hand motion, when CMV retinitis was first diagnosed. However, visual acuity testing was difficult in patients transferred to the intensive care unit during treatment.

In conclusion, our findings suggest that the visual prognosis was worse in patients with greater invasion of the central retinal zone, those with a longer interval between HSCT and the diagnosis of CMV retinitis, and those who died. CMV retinitis can lead to various complications despite active and aggressive treatment, and it can result in severe vision loss. Therefore, once CMV viremia is confirmed, we recommend an immediate fundus examination. While CMV viremia is persistent, it is repeated every 2 weeks even if there is no CMV retinitis. In addition, repeated examinations are recommended every 2 weeks for at least 2 months, even after CMV DNA titers in peripheral blood become negative.

## Supporting information

**S1 File.**
(XLSX)

## Author Contributions

**Conceptualization:** Joo Young Kim, Young-Hoon Park.

**Data curation:** Joo Young Kim, Seo-Yeon Hong.

**Formal analysis:** Joo Young Kim.

**Funding acquisition:** Young-Hoon Park.

**Investigation:** Seo-Yeon Hong, Woo Kyung Park, Rae Young Kim, Mirinae Kim.

**Methodology:** Joo Young Kim, Woo Kyung Park, Rae Young Kim, Mirinae Kim.

**Project administration:** Young-Hoon Park.

**Resources:** Young-Hoon Park.

**Software:** Joo Young Kim, Seo-Yeon Hong.

**Supervision:** Hee-Je Kim, Seok Lee, Dong Gun Lee, Young-Hoon Park.

**Validation:** Young-Hoon Park.

**Visualization:** Joo Young Kim, Young-Gun Park, Young-Hoon Park.

**Writing – original draft:** Joo Young Kim.

**Writing – review & editing:** Joo Young Kim, Mirinae Kim, Young-Gun Park, Young-Hoon Park.

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
