## [Decision Letter · Decision Letter 0]

11 Jun 2020

PONE-D-20-13198

Prognostic factors of cytomegalovirus retinitis after hematopoietic stem cell transplantation

PLOS ONE

Dear Dr. Park,

Thank you for submitting your manuscript to PLOS ONE. After careful consideration, we feel that it has merit but does not fully meet PLOS ONE’s publication criteria as it currently stands. Therefore, we invite you to submit a revised version of the manuscript that addresses the points raised during the review process.

ACADEMIC EDITOR: Please consider revising manuscript as detailed by reviewers

We look forward to receiving your revised manuscript.

Kind regards,

Shree K. Kurup, MD

Academic Editor

PLOS ONE

Journal Requirements:

Additional Editor Comments (if provided):

Please kindly VERY SPECIFICALLY address the reviewers comments

Reviewers' comments:

Reviewer's Responses to Questions

**Comments to the Author**

1. Is the manuscript technically sound, and do the data support the conclusions?

Reviewer #1: Yes

Reviewer #2: Yes

2. Has the statistical analysis been performed appropriately and rigorously? 

Reviewer #1: Yes

Reviewer #2: Yes

3. Have the authors made all data underlying the findings in their manuscript fully available?

Reviewer #1: Yes

Reviewer #2: Yes

4. Is the manuscript presented in an intelligible fashion and written in standard English?

Reviewer #1: Yes

Reviewer #2: No

5. Review Comments to the Author

Reviewer #1: The paper by Young-Hoon Park et al. presents a retrospective study where the presence of cytomegalovirus retinitis after hematopoietic stem cell transplantation was analysed in more than 4000 patients. The paper is well written and the data well documented.

Minor:

- The title does not reflect the main conclusion of the study.

- the introduction is more focused on the relation between cytomegalovirus retinitis and AIDS than on hematopoietic stem cell transplantation. This needs to be corrected to better introduce the aim of the study.

- Table 5 is quite complex making it hard to read.

Major:

- The authors should consider new scientific pieces of evidence. Please note that on the references there is only one paper from 2019 and one from 2017 being all the others previous to 2017.

Reviewer #2: This is a retrospective study of patients with the aim of knowing the visual prognostic factors in patients who presented CMV retinitis after hematopoietic cell transplantation of various etiologies between 2010 and 2019 in the Republic of Korea. Of the 4241 patients who underwent HSCT, 1063 presented CMV viremia, and of those, only 67 patients (93 eyes) presented CMV retinitis. Only one was excluded because he presented infectious endophthalmitis before the development of CMV.

They collected all demographic data of the patients, as well as the registration of the primary disease, CMV viremia, the diagnosis of retinitis concerning the time of HSCT and detected viremia, visual acuity at the beginning, follow-up, and end; affected area, the positivity of aqueous humor samples, among others.

Only 48 % of the patient exhibit sytoms, 30% of the patients die within the first year of the transplant and after the multiple regressin analysis, the 3 factor that influence the final visual acuity were involved zone, time between HSCT and retinitis diagnosis and survival status.

In conclusion a prompt retinal evaluation and frequent repeat examinations are necessary. Examinations should continue even if the CMV DNA titer in the peripheral blood becomes negative

I think this is a good piece of research on a topic that has changed due to the advances in hematopoietic stem cell transplant; also, it can have an important impact on future screening policies. Therefore, I suggest reviewing the writing of the manuscript and the following recommendations:

Abstract

Include, site, patients, and years of the retrospective series and summarize the data of the variables to be studied and, If possible, reduce the secondary objectives to be discussed in the methodology section of the manuscript.

Introduction

Line 79 Lower how much?; I suggest citing a percentage from previews reports.

Line 80-83: I suggest rephrasing this paragraph

Line 84: Word transplantation is repeated in the same sentence

Line 89-90: Passive voice misuse

Line 91-94 I Suggest re-order the paragraph, for example

¨ Previous studies have shown high-risk characteristics for the development of CMV retinitis like prolonged CMV viremia, high peak CMV DNA levels in the peripheral blood after HSCT, transplants from unrelated donors, and transplants from human leukocyte antigen (HLA)-mismatched donors.¨

Line 95-96: As I understand your main objective is the visual prognosis, but in the Abstract section you stated: ¨ To identify the prognostic factors in patients with cytomegalovirus (CMV) retinitis after hematopoietic stem cell transplantation (HSCT)¨

Just be sure to make the objective of your study is consistent between this section and the abstract

Methods

Patients Section: I suggest rewriting this section, leaving this sentence at the end of the paragraph ¨Institutional Review Board/Ethics Committee approval was obtained (Seoul Saint Mary’s Hospital), and the study was conducted in accordance with the Declaration of Helsinki. The need for informed consent was waived due to the retrospective nature of the study¨

Line 115: If there no viremia detected at baseline or weekly samples, does the patient received any other ophthalmological examination? Or only when symptoms appeared?

Line 120: After the 2-month follow-up, the patients receive another fundus examination? Or only when symptoms appeared?

Results

Line 175-177: The mean value and standard deviation of visual acuity give the idea that in your distribution is between 20/10 and 20/126 after the LogMar Snellen conversion.

¨Example: last visit 0.309 ± 0.547, mean 0.309 (20/40), -2.38 logMAR (20/11), +0.85 logMAR (20/141)¨

I suggest switch standard deveation for min or max values or range.

(Modify the table 2 with the new value)

Line 192: The 20 patients that died during the first year were included in BCVA analysis of final visual acuity? In the case that they were included in the mean final VA, How many patients did not have fundus evaluation because of systemic worsening and had ¨good¨ final visual acuity?

- When was the cutoff point for the final visual acuity?

- If one of the predictor variables for visual acuity is the survival status, I think you could show in table 2 the visual acuity of the patients that remain alive in at the end of the follow-up and in another row the visual acuity of patients that die

Line 238: Is there any relationship between the variables analyzed that correlate with the serious complications of the 17 patients described in Table 5?

Discussion

In the first paragraph describe briefly your study and the findings before you compare them with other studies

Line 253 High incidence of seropositivity in the Korean population, how much is ¨High¨? compared with other countries?

Line 260-262: The 10 patients that are mention in this section became negative for CMV DNA after the HSCT or after CMV treatment? please explain

Line 268: Same as line 120

Line 301-302: what do you mean by ¨ the changes of immune function could not be measured ¨?

Are you referring to Engraftment or the levels of neutrophils and lymphocyte after the HSCT?

Conclusion even if more information is needed to make an official screening program, what is your recommendation for the screening follow-up, taking into account that one of the visual acuity predictors is the time between the HSCT and the diagnosis.

Larochelle, M. B., Phan, R., Craddock, J., Abzug, M. J., Curtis, D., Robinson, C. C., Palestine, A. G. (2017). Cytomegalovirus Retinitis in Pediatric Stem Cell Transplants: Report of a Recent Cluster and the Development of a Screening Protocol. American Journal of Ophthalmology, 175, 8–15. doi:10.1016/j.ajo.2016.09.039

6. PLOS authors have the option to publish the peer review history of their article (what does this mean?). If published, this will include your full peer review and any attached files.

Reviewer #1: No

Reviewer #2: Yes: Juan Carlos Romo-Aguas

---

## [Author Response · Author response to Decision Letter 0]

24 Jul 2020

Reviewer #1: The paper by Young-Hoon Park et al. presents a retrospective study where the presence of cytomegalovirus retinitis after hematopoietic stem cell transplantation was analysed in more than 4000 patients. The paper is well written and the data well documented.

Minor:

- The title does not reflect the main conclusion of the study.

Response → We thank the reviewer for the valuable comment. The words in the main conclusion were changed to "visual prognosis" instead of "visual outcome," for the title to better reflect the aim of the study. (Abstract Conclusion, page 2, line 50 and Discussion, page 14, line 315)

- The introduction is more focused on the relation between cytomegalovirus retinitis and AIDS than on hematopoietic stem cell transplantation. This needs to be corrected to better introduce the aim of the study.

Response → As per your comments, in the introduction, AIDS-related content has been reduced from 2 paragraphs to 1 paragraph, and we focused on comparing HSCT recipients with other immunosuppressive conditions. (Introduction, pages 3, lines 76-83)

- Table 5 is quite complex making it hard to read.

Response → We removed a few non-critical items (type of transplantation, duration of intravitreal antiviral treatment) and edited Table 5 to make it easier to read. (Results, page 11)

Major:

- The authors should consider new scientific pieces of evidence. Please note that on the references there is only one paper from 2019 and one from 2017 being all the others previous to 2017.

Response → As per your comments, we have revised the manuscript to reflect the latest knowledge by adding 6 papers published after 2017 to the reference.

15. Peters RPH, Kestelyn PG, Zierhut M, Kempen JH. The Changing Global Epidemic of HIV and Ocular Disease. Ocul Immunol Inflamm. 2020:1-8. Epub 2020/05/13. doi: 10.1080/09273948.2020.1751214. PubMed PMID: 32396027.

Introduction, page 3, line 81

16. Diaz L, Rosales J, Rosso F, Rosales M, Estacio M, Manzi E, et al. Cytomegalovirus disease in patients with hematopoietic stem cell transplantation, experience over 8 years. Hematol Transfus Cell Ther. 2020;42(1):18-24. Epub 2019/10/19. doi: 10.1016/j.htct.2018.10.004. PubMed PMID: 31623977; PubMed Central PMCID: PMCPMC7031091.

Introduction, page 3, line 82-83

19. Yan CH, Wang Y, Mo XD, Sun YQ, Wang FR, Fu HX, et al. Incidence, risk factors, and outcomes of cytomegalovirus retinitis after haploidentical hematopoietic stem cell transplantation. Bone Marrow Transplant. 2020;55(6):1147-60. Epub 2020/01/30. doi: 10.1038/s41409-020-0790-z. PubMed PMID: 31992849.

Introduction, page 3, line 83

Discussion, page 12, line 249-251, 254-255, and page 13, line 286-288

28. Zuhair M, Smit GSA, Wallis G, Jabbar F, Smith C, Devleesschauwer B, et al. Estimation of the worldwide seroprevalence of cytomegalovirus: A systematic review and meta-analysis. Rev Med Virol. 2019;29(3):e2034. Epub 2019/02/02. doi: 10.1002/rmv.2034. PubMed PMID: 30706584.

Discussion, page 12, line 253

29. Choi SR, Kim K-R, Kim DS, Kang J-M, Kim SJ, Kim JM, et al. Changes in Cytomegalovirus Seroprevalence in Korea for 21 Years: a Single Center Study. Pediatric Infection & Vaccine 2018;12(25):123-31. doi: https://doi.org/10.14776/piv.2018.25.e8.

Discussion, page 12, line 253

32. Larochelle MB, Phan R, Craddock J, Abzug MJ, Curtis D, Robinson CC, et al. Cytomegalovirus Retinitis in Pediatric Stem Cell Transplants: Report of a Recent Cluster and the Development of a Screening Protocol. Am J Ophthalmol. 2017;175:8-15. Epub 2016/10/18. doi: 10.1016/j.ajo.2016.09.039. PubMed PMID: 27746296

Discussion, page 13, line 273-276

Reviewer #2: 

Abstract

Include, site, patients, and years of the retrospective series and summarize the data of the variables to be studied and, If possible, reduce the secondary objectives to be discussed in the methodology section of the manuscript.

Response → We thank the reviewer for the valuable comments. Accordingly, we indicated the site, patients, years, and date of this retrospective study in the abstract methods section and reduced the number of secondary objectives. (Page 2, lines 39-45)

Introduction

Line 79 Lower how much?; I suggest citing a percentage from previews reports.

Response → As per your comments, we stated that the probability of CMV retinitis occurring in HSCT recipients is 0.17-2.3%, and added references accordingly (Introduction, page 3, line 83).

Line 80-83: I suggest rephrasing this paragraph

Response → We have revised these sentences appropriately and rephrased this paragraph (Introduction, page 3, lines 76-79)

Line 84: Word transplantation is repeated in the same sentence

Response → We deleted the duplicated word, “transplantation," accordingly. (Introduction, page 3, lines 83-84)

Line 89-90: Passive voice misuse

Response → We corrected the passive voice misuse. (Introduction, page 3, lines 88-90)

Line 91-94 I Suggest re-order the paragraph, for example

¨ Previous studies have shown high-risk characteristics for the development of CMV retinitis like prolonged CMV viremia, high peak CMV DNA levels in the peripheral blood after HSCT, transplants from unrelated donors, and transplants from human leukocyte antigen (HLA)-mismatched donors.¨

Response → We revised as per your suggestions (Introduction, pages 3-4, lines 91-93)

Line 95-96: As I understand your main objective is the visual prognosis, but in the Abstract section you stated: ¨ To identify the prognostic factors in patients with cytomegalovirus (CMV) retinitis after hematopoietic stem cell transplantation (HSCT)¨

Just be sure to make the objective of your study is consistent between this section and the abstract

Response → As per your comments, we ensured consistency in the purpose in both the abstract and introduction sections by adding “visual prognostic factors.” (Abstract, page 2, line 37 and Introduction, page 4, line 95).

Methods

Patients Section: I suggest rewriting this section, leaving this sentence at the end of the paragraph ¨Institutional Review Board/Ethics Committee approval was obtained (Seoul Saint Mary’s Hospital), and the study was conducted in accordance with the Declaration of Helsinki. The need for informed consent was waived due to the retrospective nature of the study¨

Response → As per your pertinent comments, we moved the sentence to the end of the paragraph. (Methods, page 4, lines 107-109) 

Line 115: If there no viremia detected at baseline or weekly samples, does the patient received any other ophthalmological examination? Or only when symptoms appeared?

Response → Prior to HSCT, all patients undergo ophthalmological examinations, and without CMV viremia or ophthalmic symptoms, no additional examination was performed. We added this in the Methods section, page 4, lines 116-117.

Line 120: After the 2-month follow-up, the patients receive another fundus examination? Or only when symptoms appeared?

Response → If CMV retinitis does not appear, even after CMV DNA in the peripheral blood became negative and after the follow up of up to 2 months every 2 weeks, the follow up visits were gradually discontinued after increasing the intervals between examinations. Fundus examination was then performed only when CMV viremia appeared again or ophthalmic symptoms occurred. We added this in the Methods section, page 5, lines 122-124.

Results

Line 175-177: The mean value and standard deviation of visual acuity give the idea that in your distribution is between 20/10 and 20/126 after the LogMar Snellen conversion.

¨Example: last visit 0.309 ± 0.547, mean 0.309 (20/40), -2.38 logMAR (20/11), +0.85 logMAR (20/141)¨

I suggest switch standard deveation for min or max values or range.

(Modify the table 2 with the new value)

Response → As per your comments, we added the BCVA range in the Results section, page 7, lines 178-181, and page 8, Table 2. It was expressed as Snellen because no light perception (NLP) value cannot be converted to log MAR, while the hand motion (HM) can be expressed (log MAR equivalent of HM = 3) (Holladay 1997).

Holladay JT. Proper method for calculating average visual acuity. J Refract Surg. 1997;13(4):388-91. Epub 1997/07/01. PubMed PMID: 9268940.

Line 192: The 20 patients that died during the first year were included in BCVA analysis of final visual acuity? In the case that they were included in the mean final VA, How many patients did not have fundus evaluation because of systemic worsening and had ¨good¨ final visual acuity?

Response → Yes, we included all the patients in the last visit BCVA analysis except for the two patients whose final visual acuity of bilateral NLP. Seven patients (10 eyes) out of 20 patients who died were unable to undergo fundus examination due to the deterioration of their condition, and 2 (3 eyes) of 7 patients had the last BCVA of 20/40 or more measured before the condition worsened.

- When was the cutoff point for the final visual acuity?

Response → Although it varied from person to person, some patients died during the treatment of CMV retinitis, while others died after the treatment was completed (Results, page 7, line 195-196). The time of period from the last visit to patients’ death was between 3 days and 3 months. We added this in the lines 196-197.

- If one of the predictor variables for visual acuity is the survival status, I think you could show in table 2 the visual acuity of the patients that remain alive in at the end of the follow-up and in another row the visual acuity of patients that die

Response → As per your comments, we presented in Table 2 the mean, standard deviation, and range of BCVA at the last visits of the patients who died and survived (Results, page 8, Table 2)

Line 238: Is there any relationship between the variables analyzed that correlate with the serious complications of the 17 patients described in Table 5?

Response → Patients with complications such as RRD or IRU received a variety of treatments. Complications may also be important visual prognostic factors, but we believe that visual prognosis can vary depending on the treatment process of complications. For this reason, complications were excluded from the analysis in Table 4.

Discussion

In the first paragraph describe briefly your study and the findings before you compare them with other studies

Line 253 High incidence of seropositivity in the Korean population, how much is ¨High¨? compared with other countries?

Response → We added the CMV seroprevalence globally and for Korea (Discussion, page 12, line 253). In addition, in the Introduction section, since the CMV seroprevalence was duplicated (previous manuscript lines 68-69), we deleted it.

Line 260-262: The 10 patients that are mention in this section became negative for CMV DNA after the HSCT or after CMV treatment? please explain

Response → When CMV viremia is diagnosed, patients received systemic treatment with ganciclovir, foscarnet, and valganciclovir (Methods, page 5, lines 139-144). After receiving the above treatment with improved CMV viremia, 10 patients were diagnosed with CMV retinitis. We added that patients received adequate systemic treatment for CMV viremia (Discussion, page 12, line 260).

Line 268: Same as line 120

Response → We deleted “a prompt retinal evaluation and frequent repeat examinations are necessary (previous manuscript line 268)” and then changed the next sentence to the recommendation for the screening for CMV retinitis, as your suggestion. (Discussion, page 12, lines 268-271)

Line 301-302: what do you mean by ¨ the changes of immune function could not be measured ¨?

Are you referring to Engraftment or the levels of neutrophils and lymphocyte after the HSCT?

Response → We also pondered on the parameters that would best reflect the immune status and related mortality in our study.

When engraftment occurs after HSCT, the risk of infection decreases. However, engraftment syndrome or graft failure often occurs even after engraftment, and can increase the mortality rate (Chang et al. 2014).

Absolute neutrophil count (ANC) may reflect the immune function after HSCT. If ANC is very low, the immune function is poor. However, ANC can also increase when infection or physical stress occurs. When ANC is above the normal range, it cannot be said that ANC and immune function are positively correlated.

Lymphocyte count is a variable related to the overall survival rate after HSCT (Fedele et al. 2012). However, the number of lymphocytes constantly changes, and it is difficult to include this in the analysis because it is not a continuously measured value. Instead, another study (Yan et al. 2020) that analyzed the relationship between CMV retinitis and lymphocytes in HSCT recipients was additionally cited.

The sentence “Additionally, the changes of immune function could not be measured (previous manuscript line 301-302).” can be thought of as ambiguous, so we deleted it.

Chang L, Frame D, Braun T, Gatza E, Hanauer DA, Zhao S, et al. Engraftment Syndrome after Allogeneic Hematopoietic Cell Transplantation Predicts Poor Outcomes. Biol Blood Marrow Tr. 2014;20(9):1407-17. doi: 10.1016/j.bbmt.2014.05.022. PubMed PMID: WOS:000340986200022.

Fedele R, Martino M, Garreffa C, Messina G, Console G, Princi D, et al. The impact of early CD4+ lymphocyte recovery on the outcome of patients who undergo allogeneic bone marrow or peripheral blood stem cell transplantation. Blood Transfus. 2012;10(2):174-80. Epub 2012/02/18. doi: 10.2450/2012.0034-11. PubMed PMID: 22337266; PubMed Central PMCID: PMCPMC3320776.

Yan CH, Wang Y, Mo XD, Sun YQ, Wang FR, Fu HX, et al. Incidence, risk factors, and outcomes of cytomegalovirus retinitis after haploidentical hematopoietic stem cell transplantation. Bone Marrow Transplant. 2020;55(6):1147-60. Epub 2020/01/30. doi: 10.1038/s41409-020-0790-z. PubMed PMID: 31992849.

Conclusion even if more information is needed to make an official screening program, what is your recommendation for the screening follow-up, taking into account that one of the visual acuity predictors is the time between the HSCT and the diagnosis.

Larochelle, M. B., Phan, R., Craddock, J., Abzug, M. J., Curtis, D., Robinson, C. C., Palestine, A. G. (2017). Cytomegalovirus Retinitis in Pediatric Stem Cell Transplants: Report of a Recent Cluster and the Development of a Screening Protocol. American Journal of Ophthalmology, 175, 8–15. doi:10.1016/j.ajo.2016.09.039

Response → Once CMV viremia is confirmed, we recommend immediate fundus examination. While CMV viremia is persistent, it is repeated every 2 weeks even if there is no CMV retinitis. We also recommend repeated fundus examinations every 2 weeks for at least 2 months, even after CMV DNA titers in the peripheral blood become negative

We added the screening program (Abstract Conclusion, page 2, line 53-55, Discussion, page 12, lines 268-271, and page 14, lines 318-321) with reference to the above indicated article (Discussion, page 13, lines 273-276).

Response → As per your advise, we used PACE to convert figure to meet the PLOS requirements.

---

## [Decision Letter · Decision Letter 1]

13 Aug 2020

Prognostic factors of cytomegalovirus retinitis after hematopoietic stem cell transplantation

PONE-D-20-13198R1

Dear Dr. Park,

We’re pleased to inform you that your manuscript has been judged scientifically suitable for publication and will be formally accepted for publication once it meets all outstanding technical requirements.

Kind regards,

Shree K. Kurup, MD

Academic Editor

PLOS ONE

Additional Editor Comments (optional):

Reviewers' comments:

Reviewer's Responses to Questions

**Comments to the Author**

1. If the authors have adequately addressed your comments raised in a previous round of review and you feel that this manuscript is now acceptable for publication, you may indicate that here to bypass the “Comments to the Author” section, enter your conflict of interest statement in the “Confidential to Editor” section, and submit your "Accept" recommendation.

Reviewer #1: All comments have been addressed

2. Is the manuscript technically sound, and do the data support the conclusions?

Reviewer #1: Yes

3. Has the statistical analysis been performed appropriately and rigorously? 

Reviewer #1: Yes

4. Have the authors made all data underlying the findings in their manuscript fully available?

Reviewer #1: Yes

5. Is the manuscript presented in an intelligible fashion and written in standard English?

Reviewer #1: Yes

6. Review Comments to the Author

Reviewer #1: (No Response)

7. PLOS authors have the option to publish the peer review history of their article (what does this mean?). If published, this will include your full peer review and any attached files.

Reviewer #1: **Yes: **C. Henrique Alves

---

## [Editor Report · Acceptance letter]

21 Aug 2020

PONE-D-20-13198R1 

Prognostic factors of cytomegalovirus retinitis after hematopoietic stem cell transplantation 

Dear Dr. Park:

I'm pleased to inform you that your manuscript has been deemed suitable for publication in PLOS ONE. Congratulations! Your manuscript is now with our production department. 

Kind regards, 

on behalf of

Dr. Shree K. Kurup 

Academic Editor

PLOS ONE